# Therapeutic Potential of Tauroursodeoxycholic Acid for the Treatment of Osteoporosis

**DOI:** 10.3390/ijms21124274

**Published:** 2020-06-16

**Authors:** Tae-Keun Ahn, Kyoung-Tae Kim, Hari Prasad Joshi, Kwang Hwan Park, Jae Won Kyung, Un-Yong Choi, Seil Sohn, Seung-Hun Sheen, Dong-Eun Shin, Soo-Hong Lee, In-Bo Han

**Affiliations:** 1Department of Orthopedic Surgery, CHA Bundang Medical Center, School of Medicine CHA University, Seongnam-si, Gyeonggi-do 13496, Korea; ajh329@gmail.com (T.-K.A.); shinde@cha.ac.kr (D.-E.S.); 2Department of Neurosurgery, School of Medicine, Kyungpook National University, Daegu 41944, Korea; nskimkt7@gmail.com; 3Department of Neurosurgery, Kyungpook National University Hospital, Daegu 41944, Korea; 4Department of Neurosurgery, CHA University School of medicine, CHA Bundang Medical Center, Seongnam-si, Gyeonggi-do 13496, Korea; hariprasadjoshi10@gmail.com (H.P.J.); kyungjaewon88@gmail.com (J.W.K.); nschoiuy@gmail.com (U.-Y.C.); sisohn@hanmail.net (S.S.); nssheen@cha.ac.kr (S.-H.S.); 5Department of Orthopedic Surgery, Yonsei University, Severance Hospital, Seoul 03772, Korea; khpark800@gmail.com; 6Department of Medical Biotechnology, Dongguk University-Seoul, Seoul 04620, Korea

**Keywords:** osteoblast, osteoporosis, bone turnover biomarkers, matrix mineralization, bone histomorphometry

## Abstract

Tauroursodeoxycholic acid (TUDCA) is a US FDA-approved hydrophilic bile acid for the treatment of chronic cholestatic liver disease. In the present study, we investigate the effects of TUDCA on the proliferation and differentiation of osteoblasts and its therapeutic effect on a mice model of osteoporosis. Following treatment with different concentrations of TUDCA, cell viability, differentiation, and mineralization were measured. Three-month-old female C57BL/6 mice were randomly divided into three groups (*n* = 8 mice per group): (i) normal mice as the control group, (ii) ovariectomy (OVX) group (receiving phosphate-buffered saline (PBS) treatment every other day for 4 weeks), and (iii) OVX group with TUDCA (receiving TUDCA treatment every other day for 4 weeks starting 6 weeks after OVX). At 11 weeks post-surgery, serum levels of procollagen type I N-terminal propeptides (PINP) and type I collagen crosslinked C-telopeptides (CTX) were measured, and all mice were sacrificed to examine the distal femur by micro-computed tomography (CT) scans and histology. TUDCA (100 nM, 1 µM) significantly increased the proliferation and viability of osteoblasts and osteoblast differentiation and mineralization when used in vitro. Furthermore, TUDCA neutralized the detrimental effects of methylprednisolone (methylprednisolone-induced osteoblast apoptosis). In the TUDCA treatment group the PINP level was higher and the CTX level was lower, but these levels were not significantly different compared to the PBS treatment group. Micro-CT and histology showed that the TUDCA treatment group preserved more trabecular structures in the distal femur compared to the PBS treatment group. In addition, the TUDCA treatment group increased the percentage bone volume with respect to the total bone volume, bone mineral density, and mice distal femur trabeculae compared with the PBS treatment group. Taken together, our findings suggest that TUDCA may provide a favorable effect on bones and could be used for the prevention and treatment of osteoporosis.

## 1. Introduction

Osteoporosis is a metabolic bone disease associated with a lack of balance between bone resorption and formation that results in low bone mass, structural deterioration of the bone, and fragility fractures (fractures after a small trauma, such as a fall from a low height, or from no identifiable trauma). Osteoporotic fractures (especially in the vertebra or hip) lead to substantial morbidity, mortality, and socioeconomic burdens [1,2,3,4,5]. Current pharmacological options target either osteoclasts, by bone resorption inhibition (antiresorptive agents), or osteoblasts, by bone formation stimulation (osteoanabolic agents) [4,6,7,8,9]. Most recent pharmacological approaches are typically based on anti-resorptive agents such as selective estrogen receptor modulators (SERMs), bisphosphonates, and denosumab (the receptor activator of the nuclear factor-κB ligand (RANKL) inhibitor). Antiresorptive agents can suppress bone turnover and osteoclastic-mediated bone breakdown, which might lead to unwanted side effects including atypical femur fractures (bisphosphonates, denosumab), osteonecrosis of the jaw, and venous thromboembolism (SERMs). Furthermore, there have been controversies about the effect of SERMS on non-vertebral fractures [4,6,7,8]. There are currently three US FDA-approved anabolic therapies, namely, teriparatide (recombinant human parathyroid hormone, rhPTH), abaloparatide (PTH-related protein, PTHrP), and romosozumab (anti-human scleroscin antibody) [6,7,8,9,10]. Although osteoanabolic agents can reduce the prevalence of nonvertebral and vertebral fractures, there are still concerns about their safety due to a greater risk of osteosarcoma, hypercalcemia (teriparatide, abaloparatide), and cardiovascular events (romosozumab) [6,7,8,9,10]. Due to the limitations and undesirable effects associated with current pharmacological treatments, safe and effective alternatives are highly desirable to prevent and treat osteoporosis.

Tauroursodeoxycholic acid (TUDCA) is a taurine conjugate form of ursodeoxycholic acid (UDCA) and is an endogenous hydrophilic tertiary bile acid that is produced in humans at a low concentration. TUDCA is a commercially available bile acid derivative and has been used widely in treating cholestatic liver disease and cholelithiasis. In a multicenter study, the TUDCA was as safe and efficacious as ursodeoxycholic acid (UDCA) in patients with primary biliary cholangitis, and it was better for relieving symptoms than UDCA [11]. Studies have reported that TUDCA could have a crucial role in preventing apoptosis [12], suppressing adipogenesis of adipose-derived mesenchymal stem cells (MSCs) [13], promoting angiogenesis [14,15,16,17], and enhancing osteogenic differentiation of bone marrow-derived MSCs [16]. In our previous study on TUDCA in a mouse spinal fusion model, we suggested that TUDCA could induce bone formation and was superior to the recombinant human bone morphogenetic protein-2 (rhBMP-2), in terms of bone quality, because it promoted angiogenesis and increased trabecular bone thickness [18]. However, to our knowledge, it is not evident whether TUDCA has therapeutic effects on ovariectomy (OVX)-induced osteoporosis.

The present study showed that TUDCA has remarkable anti-osteoporotic effects in an OVX-induced osteoporosis mice model. To further make clear the mechanisms underlying the anti-osteoporotic effects of TUDCA, we also investigated the effects of TUDCA on osteoblasts and methylprednisolone (MP)-induced apoptosis in primary cultures of rat osteoblasts.

## 2. Results

### 2.1. Effect of TUDCA on the Viability and Proliferation of Osteoblasts

For a preliminary observation on the effect of TUDCA on osteoblast cell viability, osteoblasts were grown under different culture conditions (control, cells cultured in Dulbecco’s modified Eagle’s media (DMEM), cells cultured in DMEM with 1 mM TUDCA, cells cultured in serum-free media, and cells cultured in serum-free media with TUDCA) over different time points (72, 48, and 24 h). The viability of the osteoblasts was analyzed using a live/dead cell assay, a 3-(4,5-dimethylthiazol-2-yl)-2,5 diphenyltetrazolium bromide (MTT) assay, and CCK-8. As presented in Figure 1A,B, 1 mM TUDCA significantly increased osteoblast viability. Cell proliferation assayed with CCK-8 was higher when cultured in even serum-free media with 1 mM TUDCA compared to the control cells for 24 h, 48 h, and 72 h (Figure 1C). The preventive effect of TUDCA on 10 µM MP-induced osteoblast apoptosis was also measured. As shown in Figure 1D,E, 10 µM MP significantly induced osteoblasts apoptosis and 1 mM TUDCA treatment showed a significantly higher anti-apoptotic effect on the osteoblasts. These results suggest that TUDCA could substantially improve osteoblast cell viability and save osteoblasts from MP-induced apoptosis in vitro.

### 2.2. Effect of TUDCA on Osteoblast Differentiation

Alkaline phosphatase (ALP) activity is a widely used marker for early osteoblast differentiation. ALP activity in the TUDCA-treated groups (100 nM, 1 µM, and 1 mM) was found to be higher than that in the untreated groups (osteoblasts in basal or osteogenic media) after 5 days of incubation (Figure 2A). RUNX2 was also used as a biomarker for immature osteoblasts. The RUNX2 expression level in the TUDCA-treated group (100 nM and 1 µM) was significantly higher than that in the untreated group on days 1, 2, 3, and 5 (Figure 2B). Mineralization nodule formation was observed on day 14 after Alizarin Red-S staining. Higher mineralization was observed in the TUDCA (100 nM, 1 µM)-treated groups when compared to that in the other groups (cells cultured in basal or osteogenic media) (Figure 2C,D). No significant differences were seen between the 100 nM and 1 µM treatments. These results demonstrate that TUDCA can promote osteoblast differentiation. 

### 2.3. Effect of TUDCA on RANKL-Induced Osteoclastogenesis In Vitro 

The effects of TUDCA on osteoclast differentiation in vitro were evaluated using bone marrow-derived macrophages. Treatment of bone marrow-derived macrophages with M-CSF (30 ng/mL) alone did not activate tartrate-resistant acid phosphatase (TRAP)-positive cells, and M-CSF (30 ng/mL) plus RANKL (50 ng/mL) could induce the multinucleated cell formations (osteoclasts) (Figure 3A). RANKL has been reported to induce expression of osteoclast-specific genes, including RANK and MMP-9, by osteoclast precursor cells [5]. Therefore, to analyze the expression levels of the RANK and MMP-9 genes in the osteoclasts, bone marrow-derived macrophages were treated with RANKL (50 ng/mL) for 5 days without or with 1 mM TUDCA. The results suggest that 1 mM TUDCA increased the expression levels of RANK and MMP-9 genes, but there were no significant differences in RANK and MMP-9 expressions (Figure 3B,C), suggesting that TUDCA may increase the expression of osteoclast-specific genes and may affect osteoclast activity.

### 2.4. Effect of TUDCA on Trabecular Architecture of Femur in an Ovariectomized Mouse Model 

Six weeks after OVX, female C57BL/6 mice were intraperitoneally injected with PBS or TUDCA (200 mg/kg) every other day for a total of 4 weeks. The right distal femurs were harvested 1 week after the final injection to evaluate the trabecular femur bone architecture (Figure 4A). As shown in the representative micro-CT images (Figure 4B) and histological sections of the right distal femur (Figure 4C), the untreated OVX group (PBS treatment group) showed an apparent deterioration of the normal trabecular structure. By contrast, trabecular structures were significantly preserved in the TUDCA treatment group, suggesting that TUDCA treatment substantially suppressed the deleterious effects of OVX on bone trabecular microstructure. 

### 2.5. Effect of TUDCA on Bone Histomorphometry in an Ovariectomized Mouse Model

The histomorphometric measurements for the test groups are presented in Figure 5. All histomorphometric measurements demonstrated a significant change in the untreated OVX group (PBS treatment group) compared to the values in the control group (Figure 5). There was a decline in percentage bone volume (BV/TV, %), bone mineral density (BMD, g/cm^3^), Tb.N (mm^−1^), and Tb.Th (mm), and an increase in Tb.Sp (mm). By contrast, in the TUDCA treatment group, there were significant increases in all measured parameters compared to the values in the untreated OVX group, with the exception of Tb.Sp, which showed a significant decline (Figure 5). Accordingly, there were significant differences between the untreated OVX group and the TUDCA treatment group (Figure 5). These results demonstrate that TUDCA had highly significant positive effects on bone trabecular microstructure.

### 2.6. Effect of TUDCA on Bone Metabolism in an Ovariectomized Mouse Model

To analyze the effect of TUDCA on bone metabolism, the serum levels of a bone formation marker, PINP, and a bone resorption marker, CTX, were analyzed at 11 weeks post-OVX. The serum PINP levels were 0.81 ± 0.11 ng/mL in the control group, 0.79 ± 0.13 ng/mL in the untreated OVX group, and 0.89 ± 0.11 ng/mL in the TUDCA-treated group. The difference in the PINP levels between the untreated OVX group and the TUDCA-treated group was not statistically significant, but higher values were observed in the TUDCA-treated group than in the untreated OVX group. In the analysis of the serum CTX level, the mean values were 0.083 ± 0.016 ng/mL in the control group, 0.085 ± 0.012 ng/mL in the untreated OVX group, and 0.082 ± 0.009 ng/mL in the TUDCA-treated group. The differences in the CTX levels between the control group and the untreated OVX group and between the untreated OVX group and the TUDCA-treated group were not statistically significant, but lower values were observed in the TUDCA-treated group than in the untreated OVX group (Figure 6). The results suggest that TUDCA could inhibit bone loss at 11 weeks after OVX.

## 3. Discussion

Postmenopausal osteoporosis is associated with low bone mass that results from increased resorption of bone and decreased formation of bone, eventually leading to a greater risk of fractures [1,2,3,4,5]. Promising pharmacotherapies for postmenopausal osteoporosis have improved health conditions for all patients [6,7,8,9,10]. However, certain drugs may not be appropriate for all patients and may cause various side effects in some individuals. Thus, it is still necessary to study novel drugs that are effective and safe for the treatment of postmenopausal osteoporosis. Here, we suggested that TUDCA could promote osteoblast viability and differentiation, protect osteoblasts from MP-induced apoptosis in vitro, and clearly improve OVX-induced osteoporosis in a mouse model. To our knowledge, this is the first study that identifies the anti-osteoporotic effects of TUDCA.

In the current study, we first explored the effects of TUDCA on osteoblast viability and differentiation in vitro. RUNX2 is a critical transcription factor that is presented in the earliest stage of osteogenic differentiation, and ALP is an early marker of osteogenic differentiation [19]. Therefore, ALP activity was evaluated after 5 days of incubation, and the expression of the important osteoblast-specific gene RUNX2 was measured at 24, 48, and 72 h and 5 days of incubation. On day 5, TUDCA significantly increased ALP activity and RUNX2 expression. In addition, TUDCA at 100 nM and 1 µM concentrations improved bone mineralization and 1 mM TUDCA protected osteoblasts from MP-induced apoptosis. TUDCA has been proven to inhibit endoplasmic reticulum (ER) stress, which is associated with osteoclastogenesis [20]. An inducer of ER stress (thapsigargin) was demonstrated to decrease osteoclast formation from RAW264.7 macrophages [21]. The inhibition of ER stress could induce the downregulation of monocyte chemotactic protein-1 (MCP-1)-mediated osteoclast precursor differentiation [22]. In a recent report, Wang et al. demonstrated that the inhibition of ER stress by TUDCA significantly decreased the formation of osteoclasts medicated by TiAl6V4 alloy particles (*TiPs*) in vitro and in vivo [23]. Thus, in our study, we also analyzed the effects of TUDCA on osteoclasts and found that TUDCA did not significantly inhibit the expression of osteoclast-specific genes, such as MMP-9 and RANK. All of these in vitro results suggested that TUDCA could play an important role in promoting osteoblast viability and differentiation as well as in preventing glucocorticoid-induced osteoblast apoptosis. 

In the present study, we did not evaluate the effect of TUDCA on inflammatory reaction. It is well known that TUDCA has a powerful anti-inflammatory effect [24]. Human and animal experiments showed pro-inflammatory cytokines as primary mediators of the accelerated bone loss at post-menopause including interleukin-1, tumor necrosis factor-alpha, and interleukin-6. These pro-inflammatory cytokines are associated with osteoclastic bone resorption in various situations [25]. Therefore, the anti-inflammatory effect of TUDCA may be one of its anti-osteoporotic effects. It should be evaluated in a future study. 

Bilateral ovariectomy was performed to establish an OVX mouse model of osteoporosis. Ovariectomy-induced osteoporosis in rodents produces skeletal responses that are similar to those in post-menopausal women [26]. Both structural and mechanical investigations can be easily performed on the distal femur of rodents due to the geometry of the bone and the greater amount of tissue available for sampling. Hence, in our OVX-induced osteoporosis mice model, the distal femur was chosen to analyze the effect of TUDCA on OVX-induced osteoporosis. BMD is considered an important indicator that reflects bone metabolism status and is widely used to analyze changes in bone mass and predict fracture risk [1,2,3,4,18,24]. In addition, the levels of PINP and CTX are commonly used as biochemical markers for bone resorption and formation, respectively [1,2,3,4,27]. Thus, we analyzed the effect of TUDCA on the levels of PINP and CTX. The primary findings of our in vivo study are as follows: (i) TUDCA remarkably reversed the ovariectomy-induced reduction in BMD; (ii) TUDCA not only elevated the PINP levels but also reduced the CTX levels in an OVX-induced osteoporosis model, even though this did not result in a statistical difference due to small number of animals (8/each group); (iii) TUDCA treatment clearly preserved trabecular microstructures in vivo. Taken together, our results indicate that TUDCA could alleviate OVX-induced osteoporosis in a mice model. 

Although the underlying mechanisms implicated in the anti-osteoporotic effects of TUDCA remain unclear, we suggest several possibilities based on previous studies and our current results: (i) promotion of osteoblast survival and inhibition of osteoblast apoptosis, (ii) suppression of adipogenesis in adipose tissue-derived MSCs by modulating endoplasmic reticulum (ER) stress, (iii) promotion of angiogenesis by recruiting vasculogenic progenitor cells, (iv) promotion of osteogenic differentiation from MSCs, and (v) its osteogenic potential by inducing new bone formation [15,16,17]. Currently available bone-forming agents such as teriparatide, abaloparatide, and romosozumab have different effects on bone surfaces [6,7]. Teriparatide and abaloparatide improve bone mass by inducing greater bone formation than resorption [28]. Teriparatide stimulates osteoblast number and activity, and can stimulate RANKL production from the osteoblasts, leading to bone resorption [16,17,18]. Abaloparatide could cause greater net bone formation than resorption compared to teriparatide [9]. In contrast, romosozumab acts primarily by activating bone formation while inhibiting bone resorption [6]. In this study, we found no evidence that TUDCA inhibited the expression of osteoclast-specific genes, including MMP-9 and RANK. However, we observed that the TUDCA-treated group had higher levels of MMP-9 and RANK, even though there was no statistical significance. These findings raise the possibility that TUDCA could stimulate an increase in osteoblast number and activity as well as RANKL production from the osteoblasts, possibly leading to bone resorption. However, TUDCA induced greater net bone formation than resorption, similar to abaloparatide.

## 4. Materials and Methods

### 4.1. Drugs

The drugs used in the current study were tauroursodeoxycholic acid (TUDCA; Tokyo Chemical Industry Co., Tokyo, Japan) and methylprednisolone (MP; corticosteroid).

### 4.2. Isolation and Culture of Osteoblasts from Rat Calvaria

Rat calvaria were collected from 3-day-old Sprague-Dawley rats. All procedures involving animals were performed following the protocol approved by the Institutional Animal Care and Use Committee (IACUC) of CHA University (IACUC 190076, March, 2019). After removal of sutures and adherent mesenchymal tissues and surgical isolation from the skull, the calvaria were introduced with 5 sequential (10–15 min) digestions at 37 °C in a solution containing 0.1% dispase and 0.1% collagenase P. Cells released from the second to the fifth digestions were centrifuged, collected, and resuspended. Those cells were cultured and incubated in proliferation medium (DMEM (Gibco BRL) supplemented with 10% (*v*/*v*) fetal bovine serum (FBS, Gibco BRL) and 100 units/mL penicillin (GibcoBRL)) or osteogenic medium (DMEM supplemented with 10% (*v*/*v*) FBS, 0.2 mM ascorbic acid (Sigma, St. Louis, MO, USA)), 1% GlutaMAX (GibcoBRL), 10 mM glycerol 2-phosphate (Sigma), and 100 units/mL penicillin (GibcoBRL) in humidified air with 5% (*v*/*v*) CO_2_ at 37 °C until the cells reached confluency.

### 4.3. Assay for Osteoblast Proliferation and Viability

To determine the effect of 1 mM TUDCA on the proliferation and viability of osteoblasts, the cells were seeded at a density of 1 × 10^3^ cells/cm^2^ on cell culture plates and incubated for 24 h under the following four conditions: (i) DMEM + 10% fetal calf serum (FCS; control), (ii) DMEM + 10% FCS with 1 mM TUDCA, (iii) DMEM only (serum-free), and (iv) DMEM only (serum-free) with 1 mM TUDCA. After 24 h of TUDCA treatment (Calbiochem, San Diego, CA), the survival of the cells in each group was visualized using the Zeiss LSM510 Meta laser scanning confocal microscope (Carl Zeiss Microimaging Inc., Göttingen, Germany) equipped with an argon laser light source after staining with fluorescein diacetate/ethidium bromide (Sigma). Cell viability was determined using a Cell Counting Kit-8 (CCK-8, Dojindo, Kumamoto, Japan) following the manufacturer’s guidance. The CCK-8 assay was used on cells cultured for different durations (72, 48, and 24 h) and the cell viability was presented as a percentage of control cells.

Cell proliferation and cell viability were analyzed using an MTT assay at 3 days of cell culture under four different conditions (control, serum-free media with TUDCA, serum-free media, and basal media with TUDCA). In addition, the effect of TUDCA on MP-induced cell apoptosis was evaluated. Osteoblasts were incubated in 96-well plates, maintained in growth media at 37 °C, and treated with 1 mM MP as well as different concentrations of TUDCA (1 μm, 10 μm, 100 μm, and 1 mM) for 72 h. Absorbance at an emission wavelength of 450 nm was measured by a microplate reader (Synergy H1 Hybrid Reader, BioTek, Winooski, VT, USA). Cell viability was presented as a percentage of the control cells or the ratio of the optical densities.

### 4.4. Alkaline Phosphatase Activity Assay

The alkaline phosphatase (ALP) activity of the cells grown with different concentrations of TUDCA was measured after 5 days. The cells were washed with PBS and lysed with 0.2 mL 0.5% Triton X-100 in PBS. The lysed cells were sonicated for 60 s and then centrifuged at 15,000× *g* for 10 min at 4 °C to remove insoluble material. The supernatant was mixed with p-nitrophenol phosphate disodium (p-NPP) substrate and incubated at 25 °C for 60 min. The optical density was spectrophotometrically determined at 405 nm using a μQuant microplate reader (μQuant, Bio-Tek, Winooski, VT, USA). The ALP activity was recorded as per μg total protein for each sample.

### 4.5. Alizarin Red-S Staining

To analyze mineralized nodule formation, rat osteoblasts were grown under the following four conditions: (i) basal media, (ii) osteogenic media, (iii) basal media with 100 nM TUDCA, and (iv) basal media with 1 µM TUDCA. On incubation day 14, the cells were fixed with 4% paraformaldehyde (PFA) at room temperature and stained with Alizarin Red-S (Sigma) at room temperature for 30 min. The mineralized nodules were examined under a microscope (magnification, ×100) and the relative value of mineralization was ascertained by determining the relative intensity of staining with image processing and analysis software (ImageJ version 1.8.0; National Institutes of Health, Bethesda, MD, USA).

### 4.6. Mouse Bone Marrow Cell Isolation

Eight-week-old Balb/c male mice were obtained from Orient Bio (Orient Bio, Inc., Seongnam, Korea). The femurs and tibias were harvested into α-Minimal Essential Medium (α-MEM; Sigma) containing 10% FBS, 100 ug/mL streptomycin, and 100 IU/mL penicillin G (culture medium) after the removal of excessive tissue. Bone marrow cells were collected from the cancellous bone marrow by flushing with culture medium using a 26-gauge needle. The bone marrow cells (2 × 10^5^) were cultured in 1 mL of culture medium with a macrophage colony-stimulating factor (M-CSF, 30 ng/mL) and RANKL (50 ng/mL) without or with TUDCA. Cultures were maintained at 37 °C in a humidified atmosphere of 5% CO_2_ in air for 6 days. Adherent cells were defined as bone marrow-derived macrophages. After 6 days, real-time polymerase chain reaction (PCR) was done to evaluate the gene markers RANK and matrix metalloproteinase-9 (MMP-9).

### 4.7. Real Time PCR

Whole RNA was extracted from transfected cells using TRIzol (Invitrogen), and 2 μg of total RNA was used for cDNA synthesis with RT-PreMix (Bioneer, Daejeon, Korea). PCR was done with PCR-PreMix (Bioneer) under typical PCR conditions. The PCR cycles consisted of an initial denaturation step at 94 °C for 5 min, followed by 32 amplification cycles consisting of 30 s of denaturation at 94 °C, 30 s of annealing at 62 °C, and 1 min of extension at 72 °C. A final extension was performed at 72 °C for 10 min. PCR products were evaluated by UV irradiation on a 1.2% agarose gel stained with ethidium bromide. For quantitative real-time PCR analysis, gene-specific primers were designed to amplify runt-related transcription factor 2 (RUNX2), glyceraldehyde 3-phosphate dehydrogenase (GAPDH), RANK, and MMP-9. Primer pairs were as follows: GAPDH (5′-ACA TCG CTC AGA CAC CAT G-3, 5-TGT AGT TGA GGT CAA TGA AGG G-3), RANK (5-AAA CCT TGG ACC AAC TGC AC-3, 5-ACC ATC TTC TCC TCC CHA GT-3), and MMP-9 (5-CGA CTT TTG TGG TCT TCC CC-3, 5-TGA AGG TTT GGA ATC GAC CC-3). All amplifications were done in a final reaction mixture (20 μL) containing one final concentration of SYBR supermix (Thermo Fischer Scientific, Waltham, MA, USA), 500 nmol/L of gene-specific primers, and 1 mL of template, using the following conditions: an initial denaturation at 95 °C for 1 min, followed by 45 cycles of 56 °C for 15 s, 95 °C for 15 s, and 72 °C for 15 s, with a final extension at 72°C for 5 min. After amplification, the threshold and baseline levels for each reaction were decided using a software package provided with the PCR system (Exicycler 96; Bioneer, Daejeon, Korea). To validate the PCR results, the amplified products were separated on 1% agarose gels and visualized by ethidium bromide staining [1,2,3,5,6].

### 4.8. Experimental Animals

Three-month-old female C57BL/6 mice (*n* = 24) were obtained from Orient Bio Inc. (Seongnam, Korea). All animal experiments were undertaken according to a protocol approved by the IACUC of our institute (IACUC170005) and the Guide for the Care and Use of Laboratory Animals (National Institutes of Health, Bethesda, MD, USA). All mice were maintained at 20 °C on a 12-h light/12-h dark cycle with free access to food and water in a pathogen-free ventilated cage.

### 4.9. Surgical Procedure and Experimental Design

After 2 weeks of stabilization, 16 mice were subjected to OVX procedure. Anesthesia was initiated in a 5% gaseous isoflurane-filled holding chamber and was maintained with 3%–4% gaseous isoflurane through a nose cone. The ventral surface of the mouse was shaved and disinfected using an alcohol and betadine solution. Buprenorphine 0.05 mg/kg was given subcutaneously. An incision was made between the thorax and the hind limb on the left flank of each mouse. Blunt forceps were then used to extract the left ovary along with the surrounding fat pad. The uterine artery and oviduct were tied using 4-0 vicryl sutures (Ethicon), and the fat pad was replaced after hemostasis. The fascia and skin were closed using 4-0 vicryl sutures (Ethicon) and a skin clip, respectively. Immediately afterward, the same procedure was performed on the right flank to extract the right ovary^18^.

The mice were divided randomly into three different groups: (i) control group without any operation or drug administration (*n* = 8); (ii) untreated OVX group (with vehicle (receiving phosphate-buffered saline [PBS]) treatment) (*n* = 8); and (iii) OVX with TUDCA treatment group (*n* = 8). Six weeks after the OVX surgery, the untreated OVX group was intraperitoneally injected with 1 mL of PBS and the OVX-TUDCA group was intraperitoneally injected with 1 mL of PBS and 200 mg/kg TUDCA. All intraperitoneal injections were conducted every other day for a total of 4 weeks. Serum analyses were performed for the biomarkers of bone resorption and bone formation at 11 weeks post-surgery. Animals were sacrificed with a CO_2_ overdose 1 week after the final injection. The right distal femurs were harvested and the effect of TUDCA on bone formation was evaluated by micro-computed tomography (CT) and histologic analyses.

### 4.10. Histomorphometric Analysis Using Micro-Computed Tomography

After TUDCA injections for a total of 4 weeks, the mice were sacrificed 1 week after the final injection. The right femur bones were harvested and the samples were fixed in 4% PFA for 48 h then stored in 70% ethanol for scanning by micro-CT. The metaphysis region of the right distal femur was scanned using a high-resolution micro-CT (SkyScan 1173, Bruker MicroCT N.V., Kontich, Belgium) at an image resolution of 9 µm (55 kV and 181 mA radiation source; 0.5 mm aluminum filter). A global threshold of 60 (1.01573 g/cm^3^) was applied to all scans to obtain a physiologically accurate representation of the trabecular bone. Morphometric parameters were then determined from binarized images using direct 3D techniques (marching cubes and sphere-fitting methods). The computed parameters were the bone mineral density (BMD, g/cm^3^), percent bone volume/tissue volume (BV/TV, %), trabecular number (Tb.N, mm^−1^), trabecular thickness (Tb.Th, mm), and trabecular separation (Tb.Sp, mm).

### 4.11. Analysis of Bone Formation and Resorption Markers

Blood samples were obtained in a serum separator tube from the direct cardiac puncture when the mice were sacrificed. Sera were stored at −83 °C until analyses. Serum type I procollagen N-terminal propeptides (PINP, bone formation marker) were evaluated using a mouse enzyme immunoassay (EIA) kit (Immunodiagnostic Systems Ltd., Boldon, UK). Type I collagen crosslinked C-telopeptide (CTX, bone resorption marker) levels in the serum were measured using the Serum CrossLaps^®^ (CTX-I) ELISA kit (Immunodiagnostic Systems Ltd., Boldon, UK).

### 4.12. Histological Examinations

After micro-CT scanning, the femurs were decalcified by immersion in a decalcification solution (National Diagnostics, Atlanta, GA, USA). The tissues were placed in a dehydrated graded series of ethanol and xylene and embedded in paraffin. Axial sections (4 µum thickness) were then obtained. The sections were stained with hematoxylin and eosin (H&E) staining and Masson’s trichrome stain to analyze new bone formation. All images were examined using an Olympus BX51 microscope (Olympus Corporation, Japan) and photomicrographs were obtained using a MicroFire digital camera with PictureFrame software (Optronics, Fremont, CA, USA).

### 4.13. Statistical Analysis

Data were collected and statistically presented as mean ± standard error (S.E). One-way analysis of variance (ANOVA) was used for the analysis of quantitative values, and Tukey’s honestly significant difference post-hoc test was used for all pair-wise comparisons among the groups. The Mann–Whitney U test was also used to compare the mean values in two groups. The Kruskal–Wallis test (with Bonferroni post-hoc tests) was used to test the significance of data to compare more than two groups. The SPSS for Windows Version 18.0 statistical software (SPSS, Chicago, IL, USA) was used for all statistical analyses. Statistical significance was determined at the *p* < 0.05 level.

## 5. Conclusions

In conclusion, the present report demonstrates that TUDCA alleviates OVX-induced osteoporosis in a mice model by increasing osteoblast viability and differentiation and protecting osteoblasts against apoptosis. TUDCA could ameliorate the deleterious effects of OVX-induced osteoporosis by inducing greater net bone formation than resorption. The data presented in this study provide evidence for the anti-osteoporotic effects of TUDCA and indicate that TUDCA may serve as an effective therapeutic strategy to treat osteoporosis.

## Figures and Tables

**Figure 1 ijms-21-04274-f001:**
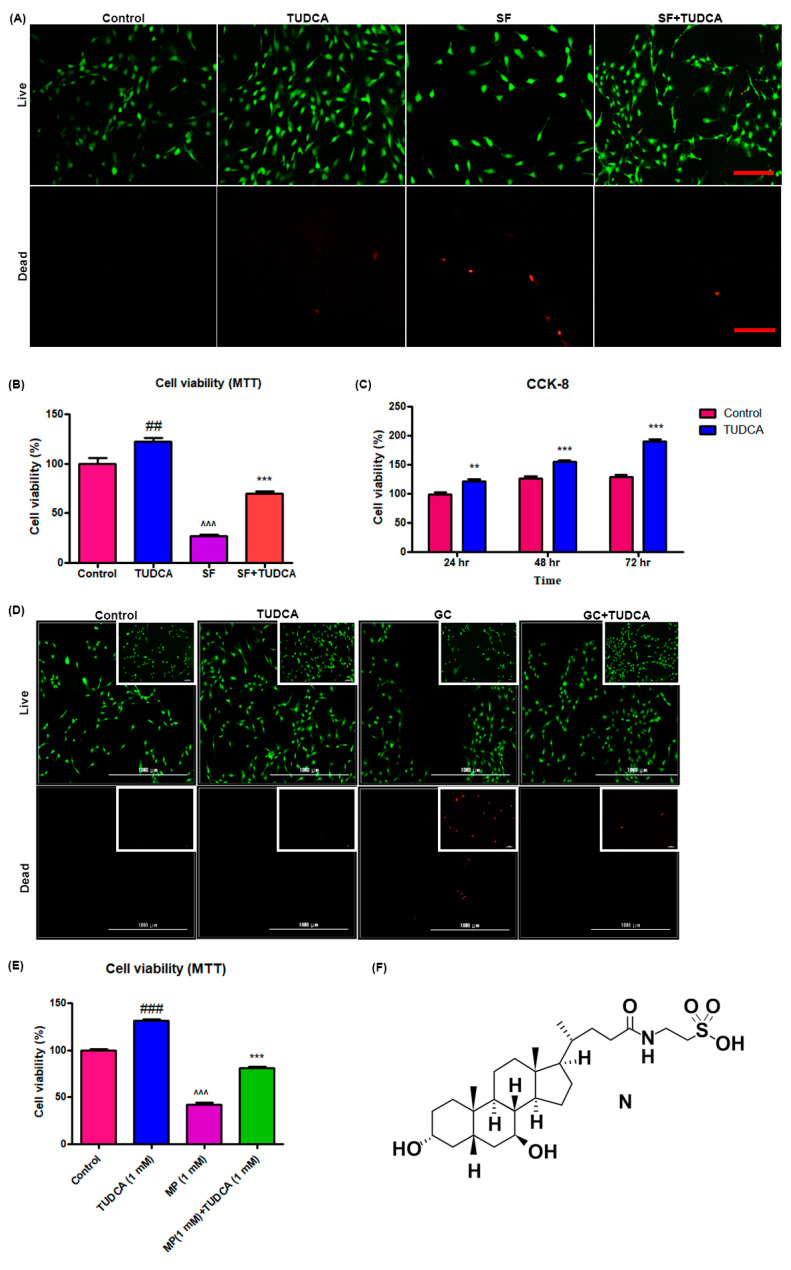
Effect of TUDCA on proliferation and viability of osteoblasts. Live/dead cell viability assay (**A**), MTT assay (**B**) (data represents means ± SEM, ^##^
*p* < 0.01 (TUDCA vs. control), ^^^^^
*p* < 0.001 (SF vs. TUDCA), *** *p* < 0.001 (SF+TUDCA vs. SF)), and CCK-8 assay (**C**) showing that 1 mM TUDCA increased viability of osteoblasts. (Data represents means ± SEM, ** *p* < 0.01, *** *p* < 0.001 (TUDCA vs. control).) Live/dead cell viability assay (**D**) and MTT assay (**E**) s+howing that TUDCA prevented methylprednisolone-induced apoptosis of osteoblasts. (**F**) Showing the chemical structure of TUDCA. Data represents means ± SEM, ^###^
*p* < 0.001 (TUDCA vs. control), ^^^^^
*p* < 0.001 (MP vs. TUDCA), *** *p* < 0.001 (MP+TUDCA vs. MP) SF: serum-free media, GC: glucocorticoid, scale bar 50 µm, 1000 µm.

**Figure 2 ijms-21-04274-f002:**
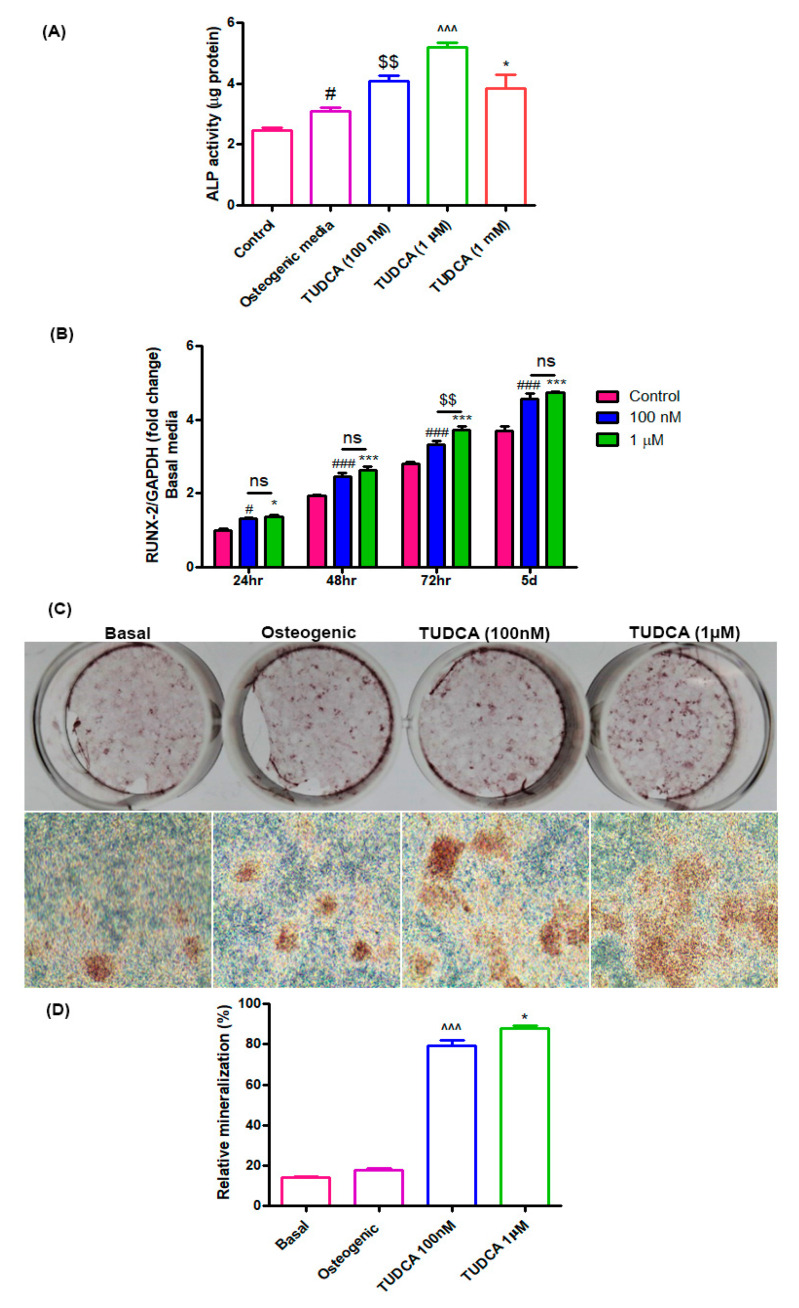
Effect of TUDCA on osteoblast differentiation. (**A**) Alkaline phosphatase activity assay for control, 100 nM, 1 µM, and 1 mM TUDCA on day 5, showing higher values in the 100 nM and 1 µM TUDCA cohort. Means ± SEM, ^#^
*p* < 0.05 (Osteogenic media vs. control), ^$$^
*p* < 0.01 (TUDCA-100 nM vs. Osteogenic media), ^^^^^
*p* < 0.001 (TUDCA-1 µM vs. TUDCA-100 nM), *** *p* < 0.001 (TUDCA-1 mM vs. TUDCA-1 µM) (**B**) The expression of RUNX2 gene increased in the 100 nM and 1 µM TUDCA cohort. Means ± SEM, ^###^
*p* < 0.001 (48 h vs. 24 h), ^^^^^
*p* < 0.001 (72 h vs. 48 h), *** *p* < 0.001 (5 d vs. 72 h) (**C**) Representative images of Alizarin Red-S staining on day 14 for different culture conditions: controls, osteogenic media, 100 nM TUDCA, and 1 µM TUDCA. Higher mineralization was observed in the TUDCA treatment cohort. Scale bar, 100 µm. (**D**) Quantitative result showing relative mineralization. Means ± SEM, ^^^^^
*p* < 0.001 (TUDCA-100 nM µ vs. Osteogenic), * *p* < 0.05 (TUDCA-1 µM vs. TUDCA-100 mM).

**Figure 3 ijms-21-04274-f003:**
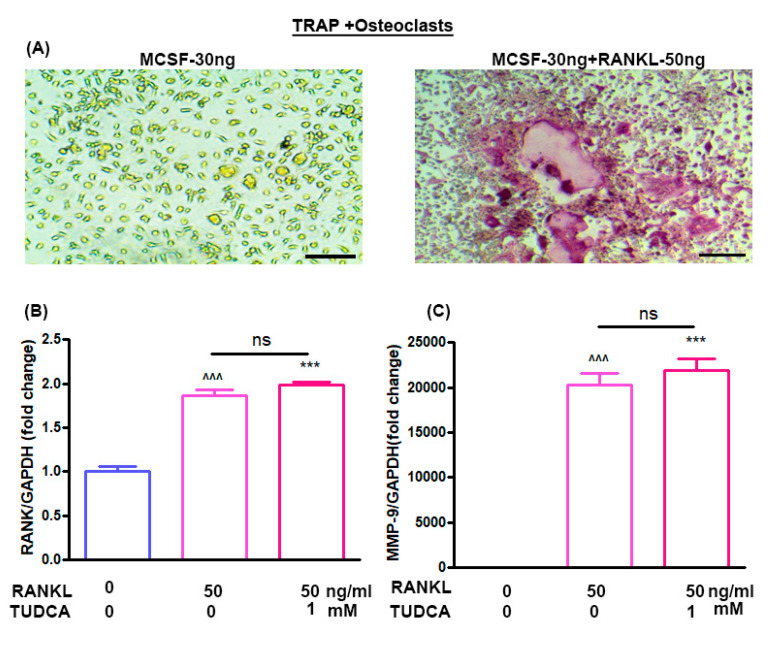
Effect of TUDCA on osteoclasts. (**A**) Stimulation of bone marrow-derived macrophages by macrophage colony-stimulating factor (M-CSF, 30 ng/mL) and receptor activator of nuclear factor kappa-B ligand (RANKL, 50 ng/mL) resulted in the formation of multinucleated cells, whose identity as osteoclast-like cells was confirmed with tartrate-resistant acid phosphatase (TRAP) staining (×200). Real-time PCR analysis of the RANK gene (**B**) and MMP-9 gene (**C**) in osteoclasts treated with or without TUDCA and RANKL. Means ± SEM, ^^^^^
*p* < 0.001 (RANKL-50+TUDCA-0) vs. (RANKL-0+TUDCA-0), *** *p* < 0.001 (RANKL-50+TUDCA-1) vs. (RANKL-0+TUDCA-0), ns; (RANKL-50+TUDCA-1) vs. (RANKL-50+TUDCA-0).

**Figure 4 ijms-21-04274-f004:**
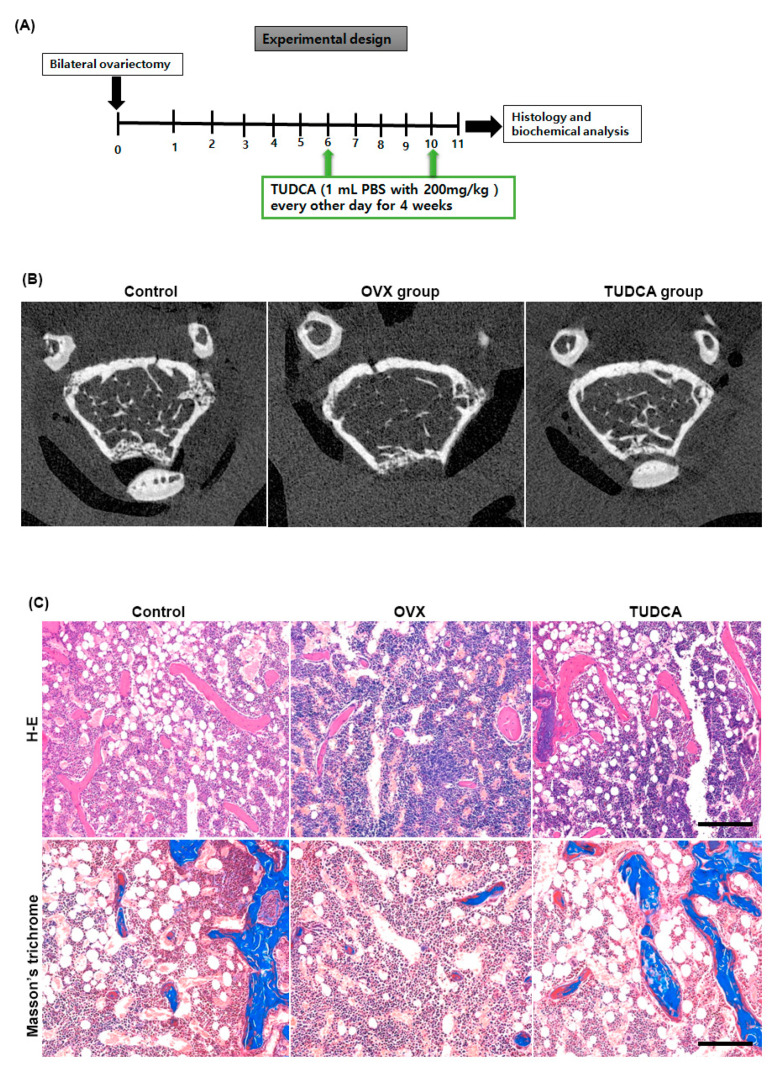
Experimental design and imaging and histology analysis. (**A**) Experimental design. (**B**) Post mortem micro-CT axial images obtained from mouse distal femur at 6 weeks after ovariectomy. The TUDCA-treated cohort showed more trabecular bone structures than the untreated OVX cohort at 6 weeks after ovariectomy. Hematoxylin and eosin (H&E) and Masson’s trichrome (**C**) staining showing increased trabecular bone formation in the TUDCA treatment cohort (×200).

**Figure 5 ijms-21-04274-f005:**
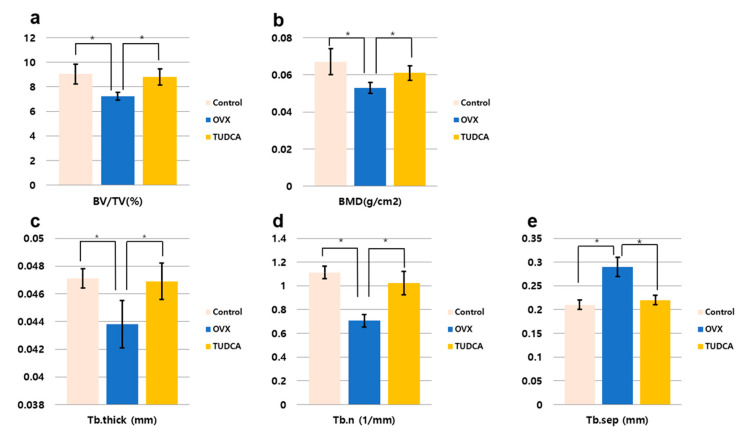
Bone histomorphometric analysis. (**a**) Percentage bone volume (BV/TV, %), (**b**) bone mineral density (BMD, g/cm^2^), (**c**) trabecular thickness (Tb.Th, mm), (**d**) trabecular number (Tb.N, 1/mm), (**e**) trabecular separation (Tb.Sp, mm), * *p* < 0.05.

**Figure 6 ijms-21-04274-f006:**
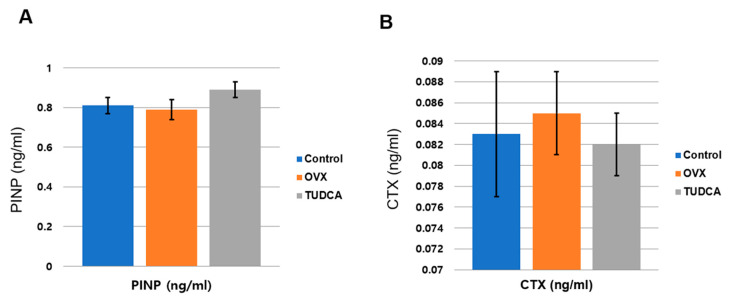
Serum bone turnover marker analysis (**A**), serum procollagen type I propeptides (PINP, ng/mL), (**B**) type I collagen crosslinked C-telopeptides (CTX, ng/mL). The differences between the control cohort and the untreated ovariectomy cohort and between the untreated ovariectomy cohort and the TUDCA-treated cohort were not statistically significant, but higher values of PINP and lower values of CTX were obtained in the TUDCA-treated cohort than in the untreated ovariectomy cohort.

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
