# Peer review of "Therapeutic Potential of Tauroursodeoxycholic Acid for the Treatment of Osteoporosis"

_ijms, 2020, doi:10.3390/ijms21124274_

Round 1
Reviewer 1 Report
In this work the therapeutic potential of TUDCA for the treatment of osteoporosis is analysed. Authors explored the viability and proliferation of osteoblasts, the differentiation, in in vitro experiments and in an ovariectomized mouse model. Results obtained sound good, with TUDCA showing protective effects in OVX-induced osteoporosis.
I think this work is well organised and developed with a good methodologic approach. Data obtained in the experiments were reported clearly and with a proper form in figures. Statistical analysis was performed to assess the significance of data. The animal experiments were carried out according to a specific protocol approved by their istitution.
I suggest to include in the manuscript a list of abbreviations, giving that there is a number of acronyms, that could make hard the reading.
Again, I suggest to insert the chemical structure of TUDCA: it could be inserted as a panel in Figure 1.
Overall, in my opinion this work is suitable for publication in this journal.
Author Response
I appreciate your favorable review. I added the list of abbreviations and chemical structure of TUDCA in Fig 1.
Reviewer 2 Report
Manuscript ID: ijms-837890
Title: "Therapeutic potential of tauroursodeoxycholic acid for the treatment of osteoporosis”
The Authors conducted both in vitro and in vivo experiments on a mice model of osteoporosis. In particular, they investigated the effects of tauroursodeoxycholic acid (TUDCA) on the sutvival, proliferation and differentiation of osteoblasts, as well as its ability to neutralize the methylprednisolone-induced osteoblast apoptosis. The effect of TUDCA on bone turnover markers (P1NP and CTX), bone mineral density (BMD) and bone structure through distal femur micro-computed tomography scans and histology have also been evaluated. and its therapeutic effect on a mice 20 model of osteoporosis. The main results show that TUDCA exerts therapeutic effect on ovariectomized mice and provide a favorable effect on bone by favouring proliferation and differentiation of osteoblasts, suggesting that it could be used for the prevention and treatment of osteoporosis.
This paper is quite interesting, rigorously structured, well written, and easy to read. Methods and Results are exhaustive, clear and detailed, containing all the necessary information. The statistical analysis applied to the data is appropriate. No major revisions are needed. I suggest only the following minors implementations:
It is well known that osteoporosis is a condition often associated to chronic liver diseases. TUDCA is an approved hydrophilic bile acid for the treatment of chronic cholestatic liver disease. Are there clinical data from the literature that highlight a possible protective effect of TUDCA in patients with chronic cholestatic liver diseases to which it is currently administered? It would be interesting to mention this in Introduction or in Discussion.
How do the Authors explain the lack of a statistically significant difference between bone turnover markers in untreated ovariectomized (OVX) group and TUDCA-treated group?
Inflammation exerts a central role in postmenopausal osteoporosis. Is it possible, based on the results, to suppose a role of TUDCA in the control of inflammation too? Could an inhibition of endoplasmic reticulum stress by TUDCA play a role in modulating inflammation and thus bone resorption?
The Authors compare the effect of TUDCA on the bone to that of PTH and abaloparatide, but the exact mechanism of action is not clear. As an anti-apoptotic agent, could TUDCA have pro-cancer effects?
In the Discussion section, it would be pertinent for the Authors to mention and better stress all these concepts.
Author Response
It is well known that osteoporosis is a condition often associated to chronic liver diseases. TUDCA is an approved hydrophilic bile acid for the treatment of chronic cholestatic liver disease. Are there clinical data from the literature that highlight a possible protective effect of TUDCA in patients with chronic cholestatic liver diseases to which it is currently administered? It would be interesting to mention this in Introduction or in Discussion.
Response
Thank you for your suggestion. I added the clinical applications in introduction.
How do the Authors explain the lack of a statistically significant difference between bone turnover markers in untreated ovariectomized (OVX) group and TUDCA-treated group?
Response
The difference in the PINP levels between the untreated OVX group and the TUDCA-treated group was not statistically significant, but higher values were observed in the TUDCA-treated group than in the untreated OVX group. The number of animal was only eight in each group, so the difference might be not significant.
Inflammation exerts a central role in postmenopausal osteoporosis. Is it possible, based on the results, to suppose a role of TUDCA in the control of inflammation too? Could an inhibition of endoplasmic reticulum stress by TUDCA play a role in modulating inflammation and thus bone resorption?
Response
I totally agree with your opinion. TUDCA has powerful anti-inflammatory effect, which influence the bone resorption. So I added below sentence in discussion
“In present study, we didn’t evaluate the effect of TUDCA on inflammatory reaction. It was well known that TUDCA has an powerful anti-inflammatory effect[24]. Human and animal experiments showed pro-inflammatory cytokines as primary mediators of the accelerated bone loss at post-menopause including interleukin-1, tumor necrosis factor-alpha, and interleukin-6. These pro-inflammatory cytokines is associated with osteoclastic bone resorption in various situation[25]. Therefore, the anti-inflammatory effect of TUDCA may be one of anti-osteoporotic effects. It should be evaluated in the future study.”
The Authors compare the effect of TUDCA on the bone to that of PTH and abaloparatide, but the exact mechanism of action is not clear. As an anti-apoptotic agent, could TUDCA have pro-cancer effects?
Response
Thank you for your excellent question. We are curious on the pro-cancer effect of TUDCA, but we could not find any report about pro-cancer effect in web search. But we could find anti-cancer effects of TUDCA in many experiments.
- Tauroursodeoxycholic acid attenuates colitis-associated colon cancer by inhibiting nuclear factor kappaB signaling. Kim YH, Kim JH, Kim BG, Lee KL, Kim JW, Koh SJ.J Gastroenterol Hepatol. 2019 Mar;34(3):544-551. doi: 10.1111/jgh.14526. Epub 2018 Nov 14.PMID: 30378164
- Tauroursodeoxycholic Acid Dampens Oncogenic Apoptosis Induced by Endoplasmic Reticulum Stress During Hepatocarcinogen Exposure. Yves-Paul Vandewynckel, Debby Laukens, Lindsey Devisscher, Annelies Paridaens, Eliene Bogaerts, et al. Oncotarget. 2015 Sep 29;6(29):28011-25. doi: 10.18632/oncotarget.4377
In the Discussion section, it would be pertinent for the Authors to mention and better stress all these concepts.
Response
Thank you very much. I did.